

**Measurement Report: Abundance and fractional solubilities of aerosol metals in urban**
**Hong Kong: Insights into factors that control aerosol metal dissolution in an urban site**
**in South China**
Junwei Yang,[1] Lan Ma,[1] Xiao He,[2] Wing Chi Au,[1] Yanhao Miao,[1] Wen-Xiong Wang,[1,3]
Theodora Nah[1,3]*
*[1]School of Energy and Environment, City University of Hong Kong, Hong Kong SAR, China*
*[2]College of Chemistry and Environmental Engineering, Shenzhen University, Shenzhen 518060, China*
*[3]State Key Laboratory of Marine Pollution, City University of Hong Kong, Hong Kong SAR, China*
*\* To whom correspondence should be addressed: Theodora Nah (Email: theodora.nah@cityu.edu.hk)*
**Abstract**
Water-soluble metals are known to produce greater adverse human health outcomes than their
water-insoluble forms. Although the concentrations of water-soluble aerosol metals are usually
limited by atmospheric processes that convert water-insoluble metals to water-soluble forms,
factors that control the solubilities of aerosol metals in different environments remain poorly
understood. In this study, we investigated the abundance and fractional solubilities of different
metals in size-fractionated aerosols collected at an urban site in Hong Kong, and identified the
factors that modulated metal solubilities in fine aerosols. The concentrations of total and water-
soluble metals in fine and coarse aerosols were the highest during the winter and spring seasons
due to the long-range transport of air masses by northly prevailing winds from emission sources
located in continental areas north of Hong Kong. The study-averaged metal fractional
solubilities spanned a wide range for both fine (8.8 % to 70.3 %) and coarse (1.4 % to 54.3 %)
aerosols, but higher fractional solubilities were typically observed for fine aerosols. Sulfate
was found to be strongly associated with both the concentrations of water-soluble Cr, Fe, Co,
Cu, Pb, and Mn and their fractional solubilities in fine aerosols, which implied that sulfate-
driven acid processing likely played an important role in the dissolution of the water-insoluble
forms for these six metals. Further analyses revealed that these strong associations were due to
sulfate providing both the acidic environment and liquid water reaction medium needed for the
acid dissolution process. Thus, the variability in the concentrations of water-soluble Cr, Fe, Co,
Cu, Pb, and Mn and their fractional solubilities were driven by both the aerosol acidity levels
and liquid water concentrations, which in turn were controlled by sulfate. These results
highlight the roles that sulfate plays in the acid dissolution of metals in fine aerosols in Hong
Kong. Our findings will likely also apply to other urban areas in South China, where sulfate is
the dominant acidic and hygroscopic component in fine aerosols.




## 1. Introduction

Chronic exposures to atmospheric aerosols, especially those in the fine mode (PM$_{2.5}$, aerosols with aerodynamic diameter $\leq$ 2.5 µm), have been linked to a myriad of deleterious effects on human health, including morbidity and excessive deaths through respiratory and cardiovascular diseases (Brook et al., 2010; Cohen et al., 2017). Some of the aerosol chemical species cause majority of the adverse human health outcomes even though they comprise a small fraction of the overall aerosol mass (Phalen, 2004; Lippmann, 2014). Metals are ubiquitous chemical species that contribute significantly to airborne aerosol toxicity even though they are typically present in aerosols in trace quantities (Costa and Dreher, 1997; Frampton et al., 1999; Ye et al., 2018; Zhao et al., 2021). Natural sources, especially mineral dust and sea spray, dominate the global sources of aerosol metals (Nriagu, 1989; Garrett, 2000; Deguillaume et al., 2005; Mahowald et al., 2018). However, anthropogenic sources such as industrial activities and vehicular traffic contribute substantial quantities of aerosol metals in urban environments (Garg et al., 2000; Adachi and Tainosho, 2004; Deguillaume et al., 2005; Lough et al., 2005; Birmili et al., 2006; Jiang et al., 2015; Mahowald et al., 2018).

Metals exist in aerosols in water-insoluble and water-soluble forms. Water-soluble metals have higher bioavailability and usually produce greater adverse human health outcomes than their water-insoluble forms (Heal et al., 2009; Fang et al., 2015; Gao et al., 2020). Some water-soluble transition metal ions (e.g., Fe(II), Fe(III), Cu(I), Cu(II)) are redox-active species and serve as catalysts in reaction cycles (e.g., Fenton-like reactions) to enhance the *in vivo* production of reactive oxygen species (ROS) (e.g., OH·, HO$_2$·, H$_2$O$_2$), which subsequently induce the physiological oxidative stress and inflammation involved in a variety of chronic and acute diseases (Bresgen and Eckl, 2015; Lakey et al., 2016; Bates et al., 2019). A recent epidemiologic study reported that water-soluble Fe concentrations in PM$_{2.5}$ showed strong correlations with cardiovascular-related emergency department visits in Atlanta (Ye et al., 2018). Less abundant water-soluble aerosol metals such as Cr and Pb are also known to exhibit both carcinogenic and noncarcinogenic risks to adults and children despite their small quantities (He et al., 2021).



Water-soluble metals also play important roles in many atmospheric processes.
Atmospheric aerosol deposition is an important source of bioavailable dissolved metals in open
oceans. The dissolved metals serve as nutrients, and in some cases toxins, for various aquatic
species (De Baar et al., 2005; Boyd et al., 2007; Paytan et al., 2009; Jordi et al., 2012). Some
transition metal ions such as Fe(III) and Mn(II) ions can facilitate the formation and aging of
organic aerosols (Chu et al., 2013; Al-Abadleh, 2015; Slikboer et al., 2015; Chu et al., 2017;
Al-Abadleh, 2021), The coupled redox cycling of Cu(I)/Cu(II) and Fe(II)/Fe(III) ions in
aerosols has been proposed to be an important mechanism for the uptake of gas-phase $HO_2$ in
aqueous aerosols, which has important implications for the tropospheric OH radical and $O_3$
budget (Mao et al., 2013; Mao et al., 2017). Mn(II)-catalyzed oxidation of $SO_2$ on aqueous
aerosol surfaces reportedly contributes more than 90 % of the sulfate production during
wintertime haze events in China (Wang et al., 2021).
Aerosol metals are primarily emitted into the atmosphere in water-insoluble forms
(Nriagu, 1989). While water-soluble aerosol metals can be emitted directly into the atmosphere
(Fang et al., 2015), the concentrations of water-soluble aerosol metals are likely limited by
atmospheric processes that convert the water-insoluble metal forms to water-soluble forms
(Mahowald et al., 2018). Given the important roles that water-soluble aerosol metals play in
adverse human health outcomes and atmospheric processes, it is necessary to understand the
factors that modulate the atmospheric processing, and hence the solubility, of aerosol metals.
Aerosol Fe dissolution has been the focus of most previous studies. A wide range (<1 % to
98 %) of fractional solubilities (ratio of the water-soluble metal mass concentration to the total
metal mass concentration) has been reported for Fe in atmospheric aerosols (Mahowald et al.,
2018). Anthropogenic-influenced aerosols generally have higher Fe solubility than fresh
mineral dust (Sedwick et al., 2007; Schroth et al., 2009; Oakes et al., 2012). However, Fe
solubility varies substantially in aerosols in different urban environments with high levels of
anthropogenic activities (e.g., 1 % to 12 % in four cities in East China (Zhu et al., 2020) vs.
around 20 % to 50 % in Hong Kong, South China (Jiang et al., 2014; Jiang et al., 2015).
Although there are a number of atmospheric processes that can influence aerosol metal
solubilities, acid processing and the formation of stable Fe-organic complexes are two key

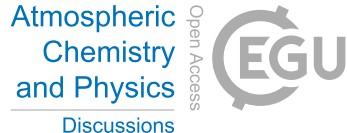

chemical processes known to enhance aerosol Fe dissolution (Deguillaume et al., 2005; Ingall
et al., 2018; Tao and Murphy, 2019; Giorio et al., 2022). At present, it remains difficult to
explain the variability of aerosol Fe solubility in urban environments since the extent to which
aerosol Fe dissolution is controlled by factors such as aerosol acidity and/or the presence of
organic ligands (e.g., oxalate) in different urban environments is still not well understood. Even
less is known about the factors that control the solubilities of other aerosol metals beyond Fe.
Hong Kong is a highly developed, densely populated city in the Guangdong-Hong
Kong-Macau Great Bay Area (GBA) urban agglomeration, which is a large business and
economic hub located in the southern part of China. While there have been some studies on the
fractional solubilities of various aerosol metals in Hong Kong (Jiang et al., 2014; Jiang et al.,
2015), to the best of our knowledge, there has not been a study that has investigated the factors
that control the solubilities of aerosol metals in Hong Kong. In this study, we investigated the
abundance and fractional solubilities of ten metals (Fe, Cu, Al, V, Cr, Mn, Co, Ni, Cd, and Pb)
in aerosols at an urban site in Hong Kong. Our main goal is to identify the key factors that
control the solubilities of metals in fine aerosols since they are believed to exert higher toxicity
than coarse aerosols due to their small sizes. We focus primarily on aerosol metal dissolution
through the acid processing and/or metal-organic complexation mechanisms. Hence, other
aerosol species were also measured for comparisons to total and water-soluble metals. The
measured aerosol inorganic ion composition was used as inputs for a thermodynamic model to
determine the aerosol acidity levels, liquid water concentrations, and pH.
**2. Methods**
**2.1. Ambient sampling**
The sampling campaign took place at ground level next to a road in Kowloon Tong
(22.3367° N, 114.1724° E). Kowloon Tong is located in the southern side of Hong Kong, and
it is primarily a residential and commercial district which is close to Mongkok, one of the
busiest commercial and most densely populated areas in Hong Kong with high density traffic
flow. Weekly size-fractionated aerosol samples were collected on 7 March 2021 to 4 April 2021
(spring season), 23 to 30 June 2021 and 7 to 14 July 2021 (summer season), 13 September



2021 to 11 October 2021 (fall season), and 15 December 2021 to 26 January 2022 (winter
season). Back-trajectories calculations calculated by the Hybrid Split-Particle Lagrangian
Integrated Trajectory (HYSPLIT) model using meteorological data from NCEP/NCAR
Reanalysis (2.5° latitude-longitude grid) showed that the sampling site was under the influence
of continental and marine air masses during the sampling periods, though the contributions of
these air masses varied with the season (Figure S1).

An eleven stage Micro-Orifice Uniform Deposit Impactor (MOUDI) (Model 110, MSP

Corp., USA) was used to collect and divide aerosols into different aerosol size bins under
ambient conditions. Aerosols were collected on prebaked 47 mm diameter quartz filters
(Tissuquartz 2500QAT-UP, Pall Corp., USA). The nominal cut points for the MOUDI eleven
impactor stages were 0.056 µm, 0.1 µm, 0.18 µm, 0.32 µm, 0.56 µm, 1.0 µm, 1.8 µm, 3.2 µm,
5.6 µm, 10 µm, and 18 µm. In the discussion below, for simplicity, we refer to aerosols
collected on impactor stages with nominal cut points 0.056 µm, 0.1 µm, 0.18 µm, 0.32 µm,
0.56 µm, 1.0 µm, and 1.8 µm as "fine aerosols", while aerosols collected on impactor stages
with nominal cut points 3.2 µm, 5.6 µm, 10 µm, and 18 µm were referred to as "coarse
aerosols". Aerosols were collected continuously for seven days (i.e., 24 hours × 7 days). This
resulted in a total of four, two, four, and six weekly sets of aerosol filter samples collected
during the spring, summer, fall, and winter seasons, respectively. After collection, the aerosol
filter samples were immediately extracted for chemical analysis.

Thermodynamic model calculations used to determine the aerosol acidity levels, liquid

water concentrations, and pH (Section 2.3) require gas-phase $NH_3$ concentrations, ambient
temperature and relative humidity (RH) as model inputs. Hence, weekly $NH_3$ measurements
were performed during each sampling period using four passive sampling devices (PSDs) and
pre-coated collection pads (PS-100 and PS-154, Ogawa & Co., Pompano Beach, FL), except
from 7 to 28 March 2021. The exposed PSD collection pads were extracted in purified
deionized water (18.2 MΩ-cm) using the protocol recommended by the manufacturer. These
aqueous extracts were subsequently analyzed by ion chromatography (Section 2.2) to
determine the average $NH_3$ concentration during the sampling period. A Vantage Vue Weather
Station (Model 6250, Davis Instruments, USA) was used to measure ambient temperature and



RH during each sampling period.

**2.2. Chemical analysis**

Each aerosol filter sample was cut into four equal pieces for chemical analysis of
different chemical components. One of the four pieces was extracted in purified deionized
water. The resulting aqueous extract was analyzed by a Total Organic Carbon (TOC) analyzer
(TOC-VCSH, Shimadzu, Japan) to determine the concentration of water-soluble organic
carbon (WSOC). The TOC analyzer has a limit of detection (LOD) of 0.5 mg L$^{-1}$. The second
filter piece was similarly extracted in purified deionized water, and then analyzed by an ion
chromatography (IC) system (Dionex ICS-1100, ThermoFisher Scientific, USA) using an
isocratic method to determine the concentrations of water-soluble anions (NO$_3^-$, SO$_4^{2-}$, Cl$^-$,
and C$_2$O$_4^{2-}$) and cations (NH$_4^+$, Na$^+$, K$^+$, Ca$^{2+}$, and Mg$^{2+}$). Anion separation was achieved using
a 4 × 250 mm anion exchange column (Dionex IonPac AS18, ThermoFisher Scientific, USA)
equipped with a 4 × 50 mm guard column (Dionex IonPac AG18, ThermoFisher Scientific,
USA). Cation separation was achieved using a 4 × 250 mm cation exchange column (Dionex
IonPac CS12A, ThermoFisher Scientific, USA) equipped with a 4 × 50 mm guard column
(Dionex IonPac CG12A, ThermoFisher Scientific, USA). 16 mM potassium hydroxide and 31
mM methanesulfonic acid were used as eluents at a flowrate of 1.0 mL min$^{-1}$ for the anion and
cation separations, respectively. The cation IC method was also used to analyze the aqueous
extracts from the exposed PSD collection pads to determine the average NH$_3$ concentration
during each sampling week. The LODs for the cation IC method were 0.025 mg L$^{-1}$ for NH$_4^+$,
Na$^+$, and Mg$^{2+}$, and 0.025 mg L$^{-1}$ for K$^+$ and Ca$^{2+}$. The LODs for the anion IC method were
0.125 mg L$^{-1}$ for NO$_3^-$, SO$_4^{2-}$, and C$_2$O$_4^{2-}$, and 0.025 mg L$^{-1}$ for Cl$^-$.
The remaining two filter pieces were used for metal analysis. One filter piece was
extracted with purified deionized water in metal-free centrifuge tubes via sonication (1 hour),
followed by high speed vortexing (15 minutes). The resulting aqueous extract was then
acidified with 2 % HNO$_3$ prior to storage at 4 °C before chemical analysis of water-soluble
metals. The last filter piece was extracted via acid digestion for chemical analysis of total
metals. The acid digestion protocol we employed was adapted from published protocols (Jiang



et al., 2014; Jiang et al., 2015). The filter piece was extracted in an acid digestion matrix (16 N
HNO$_3$ and 12 N HCl at a 3:1 volume ratio) placed in a glass microwave vial using a microwave
synthesizer (Initiator+, Biotage, Sweden). The microwave synthesizer's digestion temperature
was ramped up to 150 °C, and then held for 15 min. This was followed by cooling and
ventilation for 30 minutes. An evaporation and recovery treatment was next performed to
remove Cl$^-$ from the matrix to reduce its interference during chemical analysis. The digestion
solution was heated to 200 °C on a hotplate. Once the solution was observed to be almost dry,
16 N HNO$_3$ was added to the solution. When the solution was observed to be almost dry the
second time, 2 % HNO$_3$ was added to the solution. The resulting solution was stored at 4 °C
before chemical analysis of total metals. A standard reference material of San Joaquin soil
(SRM 2709a, NIST) was digested and analyzed using the same protocols to evaluate the metal
recoveries. Recoveries of 59.4 % for Cr, 67.0 % for Al, 93.7 % for Fe, 93.6 % for Ni, 100.2 %
for Co, 98.6 % for Pb, 95.8 % for Cu, 99.6 % for Mn, 70.5 % for V, and 94.3 % for Cd were
observed.

The concentrations of ten water-soluble and total metals ($^{27}$Al, $^{51}$V, $^{52}$Cr, $^{55}$Mn, $^{57}$Fe,

$^{59}$Co, $^{60}$Ni, $^{65}$Cu, $^{111}$Cd, and $^{208}$Pb) were determined by an Inductively Coupled Plasma–Mass
Spectrometry (ICP–MS) instrument (NexION 1000, PerkinElmer Inc., USA). The following
parameters were used for the ICP-MS instrument: 0.98 L min$^{-1}$ nebulizer gas flow, 1.2 L min$^{-1}$
auxiliary gas flow, 15 L min$^{-1}$ plasma gas flow, 5 mL min$^{-1}$ He gas flow, 1600 W RF power,
35 rpm nebulizer pump rate, and 35 rpm sample pump rate. A multi-elemental calibration
standard (IV-STOCK-13, Inorganic Ventures, USA) was used to quantify the ten water-soluble
and total metals. An internal standard solution of $^{115}$In (10 µg L$^{-1}$) was added to all samples
and standards to monitor analytical drift. The LODs for $^{27}$Al, $^{51}$V, $^{52}$Cr, $^{55}$Mn, $^{57}$Fe, $^{59}$Co, $^{60}$Ni,
$^{65}$Cu, $^{111}$Cd, and $^{208}$Pb were 87 ng L$^{-1}$, 0.8 ng L$^{-1}$, 2.8 ng L$^{-1}$, 1.6 ng L$^{-1}$, 277 ng L$^{-1}$, 0.7 ng L$^{-1}$,
4.6 ng L$^{-1}$, 6.7 ng L$^{-1}$, 1 ng L$^{-1}$, and 0.4 ng L$^{-1}$, respectively. To identify the major sources of the
aerosol metals, source apportionment was performed with positive matrix factorization (PMF)
(Paatero and Tapper, 1994; Paatero, 1997) using the aerosol chemical components measured
by the ICP-MS and IC. Details of the PMF method used can be found in Section S1 (SI).
**2.3. Thermodynamic modeling**



The thermodynamic model ISORROPIA-II was used to determine aerosol acidity levels,
liquid water concentrations, and pH (Fountoukis and Nenes, 2007). Similar to the methodology
employed by Fang et al. (2017), we ran ISORROPIA-II for each of the MOUDI impactor stages
that collected fine aerosols. The measured water-soluble $NH_4^+$, $SO_4^{2-}$, $NO_3^-$, $Cl^-$, $Na^+$, $Ca^{2+}$, $K^+$,
and $Mg^{2+}$ ions for the aerosols collected on the MOUDI impactor stage, gas-phase $NH_3$,
ambient temperature and RH were used as model inputs. Since gas-phase $NH_3$ measurements
were not available from 7 to 28 March 2021, we used $NH_3$ measurements from 28 March to 4
April 2021 as model inputs for the spring calculations. The measured $NH_3$ concentrations
during the study ranged from 3.60 µg m$^{-3}$ to 8.18 µg m$^{-3}$, with a study-averaged concentration
of 5.01 ± 1.25 µg m$^{-3}$. ISORROPIA-II was run in "forward" mode and under the assumption
that the aerosols existed in a "metastable" equilibrium state (i.e., the aerosols only existed in
liquid form). These calculations assumed that the aerosols were in thermodynamic equilibrium
with the gas phase. While fine aerosols satisfy this equilibrium condition, equilibrium between
the gas and aerosol phases of coarse aerosols cannot be achieved due to kinetic limitations
(Fountoukis et al., 2009). Thus, aerosol pH values were not calculated for coarse aerosols.
Fine aerosol pH values were calculated based on the molal definition (Pye et al., 2020):
$$pH = -\log_{10} H_{aq}^+ = -\log_{10} \frac{1000 H_{air}^+}{W_i + W_o} \cong -\log_{10} \frac{1000 H_{air}^+}{W_i} \qquad (1)$$

where $H_{air}^+$ is the hydronium ion concentration per volume of air (µg m$^{-3}$), and $W_i$ and $W_o$ are
the aerosol liquid water concentrations (µg m$^{-3}$) associated with inorganic and organic species,
respectively. $H_{air}^+$ and $W_i$ are the outputs provided by ISORROPIA-II. $W_o$ can be estimated
from the WSOC measurements using the approach described in Section S2 (SI). WSOC
concentrations in the size-fractionated aerosols ranged from 0 to 4.6 µg m$^{-3}$. The inclusion of
$W_o$ into calculations did not impact aerosol pH significantly (Figure S2). Thus, only aerosol
pH values calculated using $W_i$ will be reported here. Similar to Fang et al. (2017), lower pH
values were typically calculated for aerosols collected on MOUDI impactor stages with smaller
nominal cut points (i.e., these aerosols had smaller aerodynamic aerosol diameters) due to the
higher mass concentrations of sulfate in these smaller aerosols. The fine aerosols were mostly
acidic, with about 74 % of the calculated pH values lying between 2 and 4.



## 3. Results and discussion

### 3.1. Total metals

Figure 1 shows the seasonal average mass concentrations of the ten measured total metals in size-fractionated aerosols. The size distributions of five of the metals (Al, Fe, Mn, V, and Cd) consistently exhibited a single mode. The modes for Mn, V, and Cd were predominantly found in the fine mode, while the modes for Fe and Al were predominantly found in the coarse aerosol mode. Figure 2a shows the seasonal average concentrations of the ten measured total metals in fine and coarse aerosols. For most of the metals, higher mass concentrations were measured during the winter and/or spring seasons. This could be attributed to the long-range transport of polluted air masses by northly prevailing winds from emission sources located in continental areas north of Hong Kong (Figure S1). The metals could be arranged in the following order based on their abundances: Fe > Al > Cu > Mn > Pb > Ni > Cr > V > Co > Cd. This order of abundance was the same for both fine and coarse aerosols.

The mass concentrations of the two most abundant metals, Fe and Al, were usually higher than 10 ng m$^{-3}$ in both fine and coarse aerosols. Fe, Al, and Cu had substantially higher mass concentrations in coarse aerosols than in fine aerosols. These three metals are known to originate mainly from dust sources (e.g., mineral dust and road dust) (Hopke et al., 1980; Garg et al., 2000; Adachi and Tainosho, 2004; Lough et al., 2005; Chow et al., 2022). In contrast, Mn, Ni, V, and Cd had higher mass concentrations in fine aerosols than in coarse aerosols. These four metals are known to be consistently found in aerosols from anthropogenic sources such as vehicle and ship emissions, combustion and industrial processes (Chow et al., 2022). Pb, Cr, and Co had mostly similar concentrations in the fine and coarse aerosols.



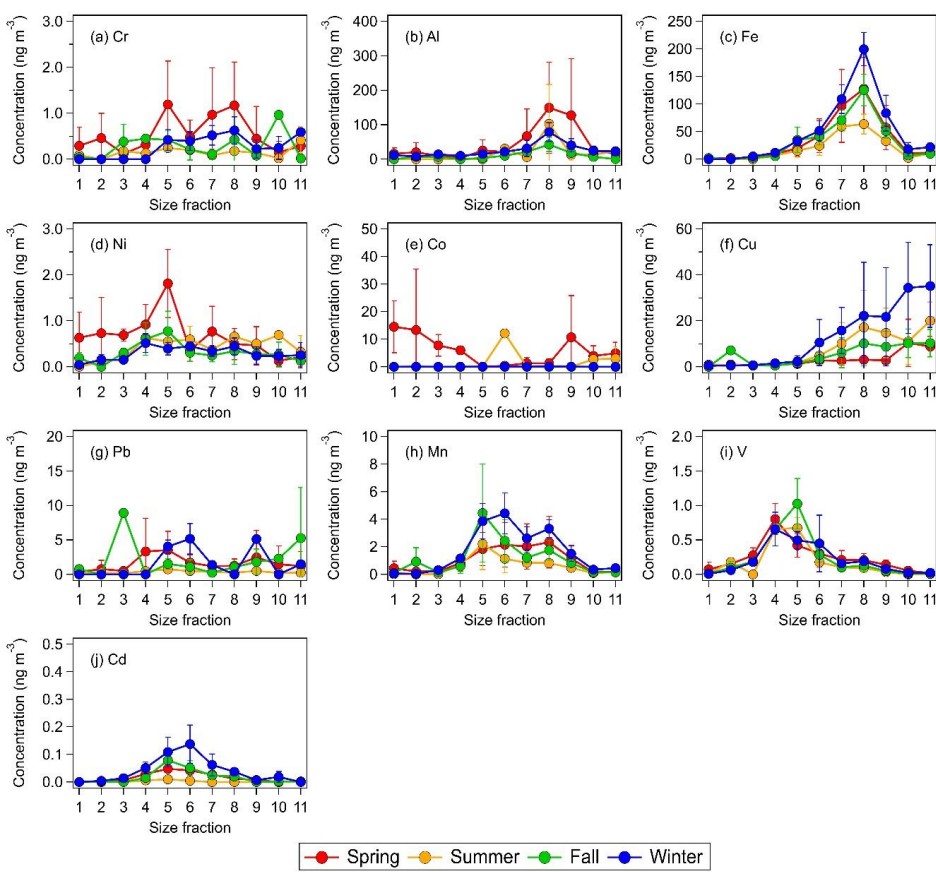

**Figure 1:** Seasonal average concentrations of total elemental metals in size-fractionated

aerosols sampled by the MOUDI with the following nominal cut points: 0.056 µm (size fraction

1), 0.1 µm (size fraction 2), 0.18 µm (size fraction 3), 0.32 µm (size fraction 4), 0.56 µm (size

fraction 5), 1.0 µm (size fraction 6), 1.8 µm (size fraction 7), 3.2 µm (size fraction 8), 5.6 µm

(size fraction 9), 10 µm (size fraction 10), and 18 µm (size fraction 11). The error bars represent

one standard deviation of the seasonal average value.


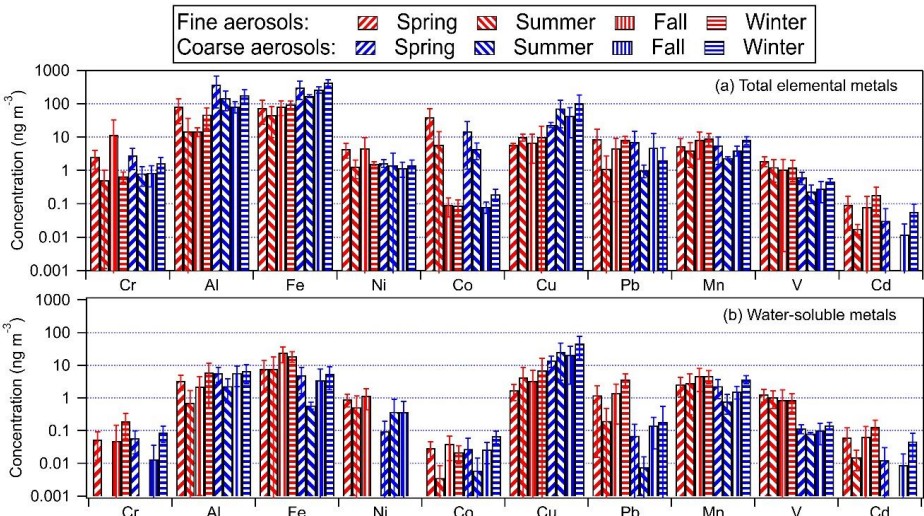

**Figure 2:** Seasonal average mass concentrations of (a) total metals and (b) water-soluble metals in fine (red) and coarse (yellow) aerosols. The error bars represent one standard deviation. The y axes are on logarithm scales.

Jiang et al. (2015) previously measured the mass concentrations of various total metals in $PM_{2.5}$ and $PM_{2.5-10}$ in Kowloon Tong. The authors carried out their measurements from 12 November 2012 to 10 December 2012 (winter) and from 8 April 2013 to 13 May 2013 (spring/summer). To gain some insights into how the aerosol metal concentrations at this urban site have changed since 2012/2013, we compared the average mass concentrations of total metals in fine and coarse aerosols measured in this study to those measured by Jiang et al. (2015). As shown in Table S1, lower mass concentrations were measured in fine (30 % to 94 % lower) and coarse (7 % to 80 % lower) aerosols for most of the metals in this study. While the lower aerosol metal mass concentrations could be partly attributed to lower levels of anthropogenic activities in 2021/2022 due to COVID-19, it is likely that the implementation of numerous local and regional air pollution policies to reduce industrial and transport-related emissions over the last decade contributed largely to this decrease. For instance, industrial upgrades resulting from the implementation of the "double transfer" policy (industry and labor transfer away from primary industries) in Guangdong likely caused the decline in the mass concentrations of metals that are typically associated with industrial activities such as Cu and





Mn (Zhong et al., 2013; Chow et al., 2022). In addition, government policies driving the switch
to cleaner fuels for energy generation and transport in Hong Kong and the GBA likely caused
the decline in the mass concentrations of metals such as Pb, Ni, V, and Fe. Interestingly, higher
mass concentrations were measured for Fe and Cu in coarse aerosols in this study compared to
those measured by Jiang et al. (2015). Fe and Cu in coarse aerosols have previously been linked
to resuspended road dust from brake and tire wear (Garg et al., 2000; Adachi and Tainosho,
2004; Lough et al., 2005). Based on publicly available government data (www.td.gov.hk), the
number of registered motor vehicles in Hong Kong has increased by about 34 % over the last
decade. It is possible that the higher Fe and Cu mass concentrations in coarse aerosols in this
study were due to increased contributions from road dust as a result of increased vehicle fleet
size at the urban site.

A PMF source apportionment analysis was performed to determine the major sources

of aerosol metals measured in this study (Section S1). A five-factor solution was selected since
it gave the most reasonable factor profiles and had high stability. The five factors were
identified as "sea salt", "dust", "ship emissions", "industrial factor 1", and "industrial factor 2"
based on the tracer species with the highest mass loadings in each factor (Figure S3). Figure
S4 shows the seasonal mass contributions of each source to each metal species in coarse and
fine aerosols. Metals with large fractions in the dust and sea salt source factor profiles generally
had higher mass concentrations in coarse aerosols. Conversely, metals with large fractions in
the ship emissions and industrial source factor profiles generally had higher mass
concentrations in fine aerosols. Higher mass contributions were usually observed in the winter
and/or spring seasons, which could be attributed to the long-range transport of polluted air
masses by northly prevailing winds from emission sources located in continental areas north
of Hong Kong (Figure S1).
**3.2. Water-soluble metals**

Figure 3 shows the seasonal average mass concentrations of water-soluble metals in

size-fractionated aerosols. The size distribution of six of the water-soluble metals (Cr, Fe, Pb,
Mn, V, and Cd) mostly exhibited a single mode, all of which were found in the fine aerosol





mode. Fe, Mn, V, and Cd exhibited a single mode for both their total and water-soluble
components (Figures 1 and 3). Of these four metals, only the modes of total and water-soluble
Fe showed obvious differences, with total Fe exhibiting a mode at around 3.2 µm (size fraction
8) and water-soluble Fe exhibiting a mode at around 0.56 µm to 1.0 µm (size fractions 5 to 6).
The modes of total and water-soluble Cu also showed obvious differences. While the mode of
total Cu was at ≥18 µm (Figure 1f), the modes of water-soluble Cu were found at substantially
smaller aerosol sizes (Figure 3f).

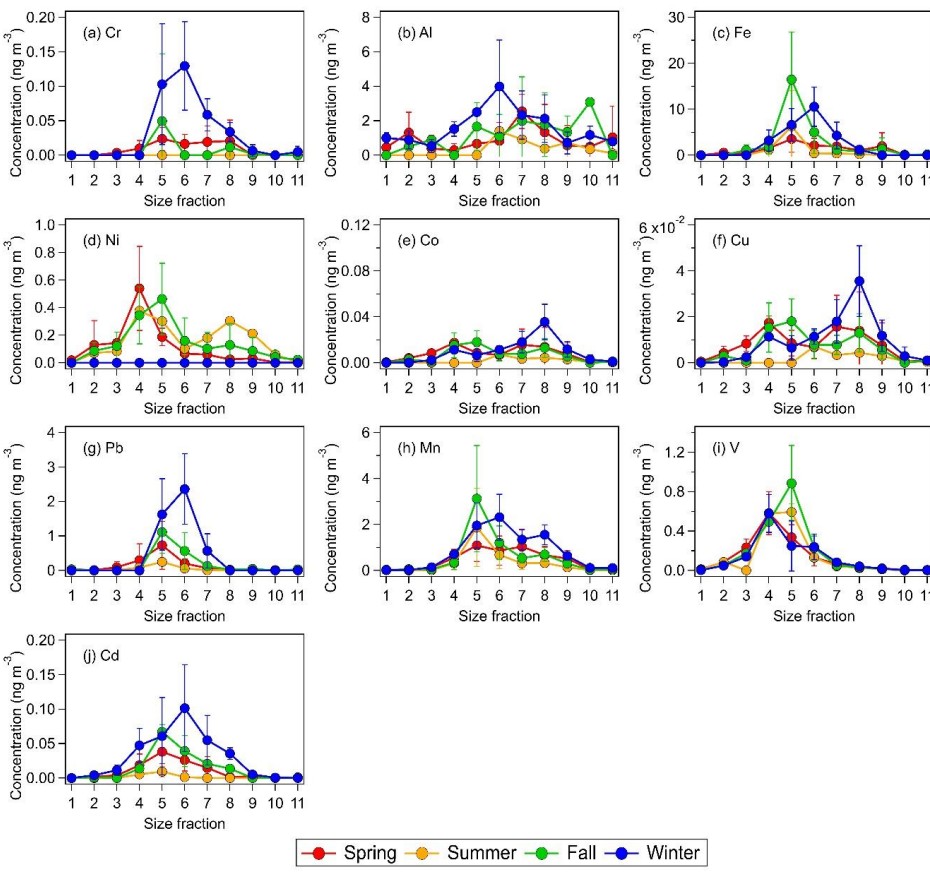


**Figure 3:** Seasonal average concentrations of water-soluble metals in size-fractionated aerosols
sampled by the MOUDI with the following nominal cut points: 0.056 µm (size fraction 1), 0.1
µm (size fraction 2), 0.18 µm (size fraction 3), 0.32 µm (size fraction 4), 0.56 µm (size fraction
5), 1.0 µm (size fraction 6), 1.8 µm (size fraction 7), 3.2 µm (size fraction 8), 5.6 µm (size

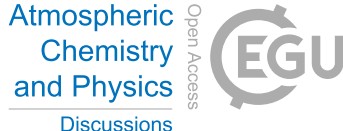

fraction 9), 10 µm (size fraction 10), and 18 µm (size fraction 11). The error bars represent one
standard deviation of the seasonal average value.

Figure 2b shows the seasonal average mass concentrations of water-soluble metals in

fine and coarse aerosols. Similar to the total metals, higher mass concentrations of water-
soluble metals were usually measured during the winter and/or spring seasons. With the
exception of Cu, the water-soluble metals usually had higher mass concentrations in fine
aerosols than in coarse aerosols. The water-soluble metals generally had the same order of
abundance as the total metals with some slight variations. The mass concentrations of water-
soluble metals generally correlated with the mass concentrations of total metals (Table S2).
This indicated that the water-soluble metals were largely derived from their total metals
through atmospheric processing, and/or that water-soluble and water-insoluble metals have the
same emission sources. For most of the metals, correlations between the mass concentrations
of water-soluble and total metals were higher for fine aerosols than for coarse aerosols. This
could be due to enhanced metal dissolution in fine aerosols via acid processing and/or the
formation of stable metal-organic complexes, which are two atmospheric chemical processes
that play key roles in influencing the solubilities of aerosol metals in many locations. This is
because acidic inorganic species that promote acid processing and organic species that can
serve as organic ligands are typically present in larger quantities in fine aerosols than in coarse
aerosols. It is also possible that differences in metal mineralogy and atmospheric processing
mechanisms in fine vs. coarse aerosols could have contributed to differences in the metal
dissolution rates (Oakes et al., 2012; Longo et al., 2016; Ingall et al., 2018).

Figure 4 shows the study-averaged fractional solubilities for the ten metals in fine and

coarse aerosols. The study-averaged metal fractional solubilities spanned a wide range for both
fine (8.8 % to 70.3 %) and coarse (1.4 % to 54.3 %) aerosols. With the exception of Cu and Co,
the metals generally exhibited higher fractional solubilities in fine aerosols compared to coarse
aerosols. The aerosol size-dependent metal fractional solubility could be explained by
differences in the aerosol composition and metal mineralogy, which resulted in different metal
dissolution rates and/or mechanisms for aerosols of different sizes. Our observations of mostly
higher metal fractional solubilities in fine aerosols are consistent with previous studies



conducted in Hong Kong (Jiang et al., 2014; Jiang et al., 2015). No season-dependent trend
was observed for the metal fractional solubilities.

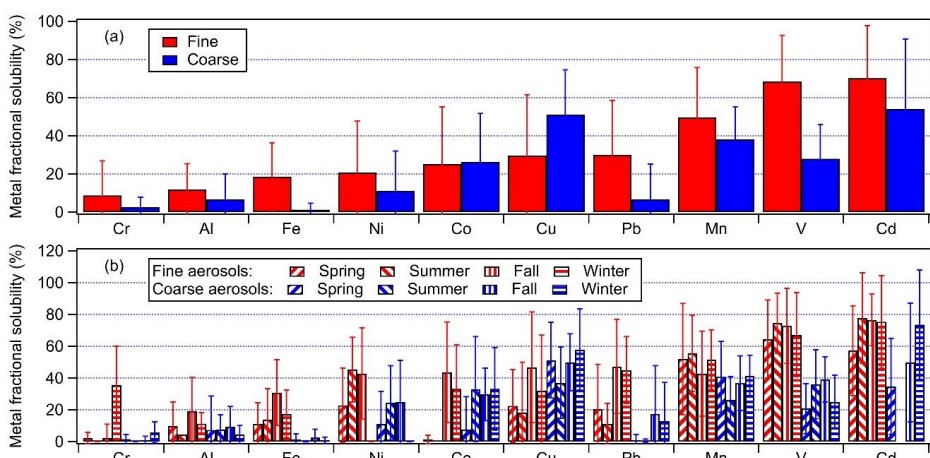

**Figure 4:** (a) Study-averaged fractional solubilities of metals in fine and coarse aerosols. (b)
Seasonal average fractional solubilities of metals in fine and coarse aerosols. The error bars
represent one standard deviation.
Some studies have reported that aerosol metal fractional solubilities will exhibit inverse
relationships with the total metal concentrations as a result of atmospheric processing (Baker
and Jickells, 2006; Sholkovitz et al., 2012; Mahowald et al., 2018; Shelley et al., 2018; Zhang
et al., 2022). There was significant scatter in many of our datasets (Figure S5), which made it
difficult to discern some of the relationships between the metal fractional solubilities and total
metal concentrations. Inverse relationships between the fractional solubility and total metal
concentration were noticeable for Cr, Al, Fe, Ni, Cu, Pb, and Mn. However, inverse
relationships between the Co, V, and Cd fractional solubilities and their total metal
concentrations were less noticeable due to their low concentrations and scatter in their datasets.
A number of factors could have contributed to the scatter in the datasets. For instance, the
scatter could be a result of the total and water-soluble metal concentrations being substantially
different in individual aerosol particles, which would not be captured by the bulk chemical
analysis performed in this study (Oakes et al., 2012; Longo et al., 2016; Ingall et al., 2018).
The metal dissolution rates in individual aerosol particles could also be significantly different





due to differences in metal mineralogy, aerosol acidity levels, presence of organic ligands etc.
in individual aerosol particles.

### 3.3. Factors that control the aerosol metal solubilities

Here, we identify the factors that control metal solubilities in fine aerosols since they
are believed to exert higher toxicity than coarse aerosols due to their small sizes. Our analyses
focus on aerosol metal dissolution via metal-organic complexation reactions and acid
processing, which are two atmospheric chemical processes believed to drive aerosol metal
dissolution in most environments. Laboratory studies have shown that the presence of organic
ligands enhances Fe dissolution in aerosols (Paris et al., 2011; Chen and Grassian, 2013; Paris
and Desboeufs, 2013; Wang et al., 2017). Water-soluble dicarboxylic acids, especially oxalate,
form stable complexes with Fe ions, which will lower the energy barrier for Fe dissolution.
While evidence of organic ligand-promoted metal dissolution in ambient aerosols has been less
conclusive, recent field studies compared the oxalate and water-soluble Fe concentrations to
show that the presence of organic ligands could contribute to aerosol Fe solubility. For instance,
strong positive correlations between oxalate and water-soluble Fe mass concentrations were
observed for $PM_{2.5}$ collected at six urban and rural sites in Canada (Tao and Murphy, 2019).
The Fe fractional solubility was also observed to be positively correlated with the molar ratio
of oxalate and Fe for $PM_{2.5}$ collected at a suburban site in Qingdao, China (Zhang et al., 2022).
To investigate whether organic ligands influenced aerosol metal solubilities in this study,
we attempted to measure oxalate in the size-fractionated aerosol samples using IC. However,
we could not detect oxalate, which indicated that the concentrations of oxalate (if present) were
below the detection limits of our IC instrument. Even though oxalate was not detected in our
size-fractionated aerosol samples, the possibility that organic ligand-promoted dissolution
contributed partly to the aerosol metal solubilities cannot be discounted completely. Oxalate
concentrations of up to about 0.5 $\mu g\ m^{-3}$ have previously been reported in $PM_{2.5}$ in Hong Kong.
In addition, a recent study reported the copresence of Fe and oxalate in individual aerosol
particles at a suburban site in Hong Kong using single particle mass spectrometry (Zhou et al.,
2020). However, organic ligand-promoted metal dissolution is a slow process, and it plays a





402 minor role in metal dissolution under low pH conditions (Zhu et al., 1993). The fine aerosols

403 collected in this study were mostly acidic, with about 60 % of the calculated pH values being

404 less than 3. This suggested that organic ligand-promoted dissolution may have played a minor

405 role in enhancing aerosol metal solubilities in this study due to the acidic nature of the aerosols.

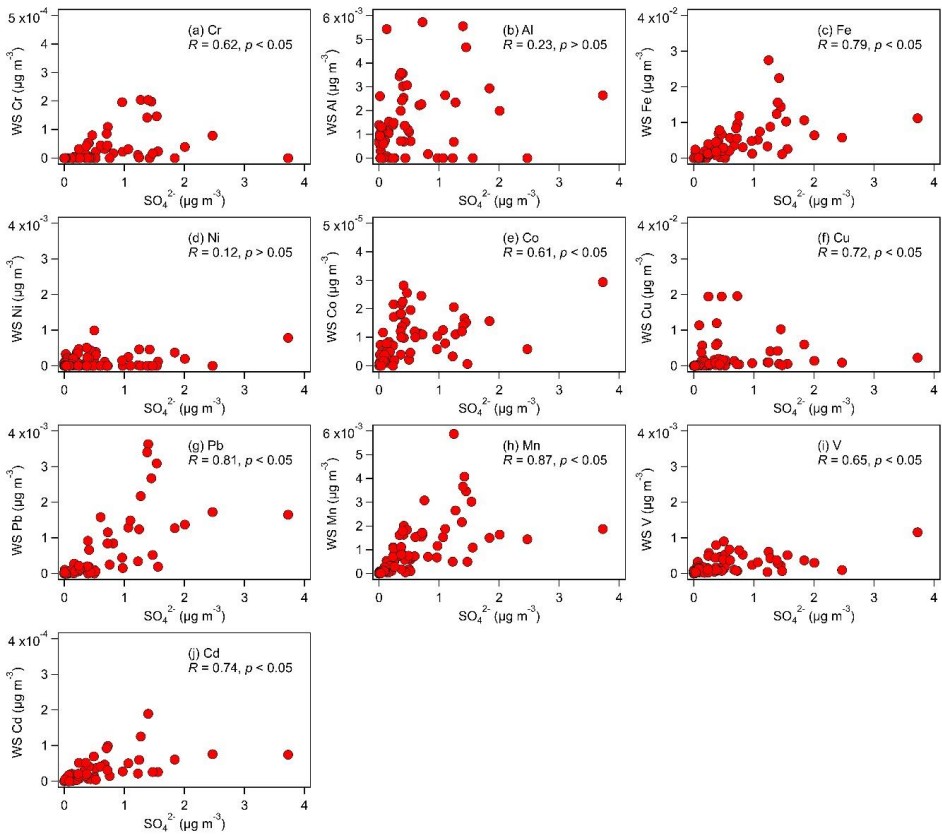

407 **Figure 5:** Relationships between the mass concentrations of water-soluble (WS) metals and

408 sulfate in fine aerosols. Only data with non-zero total metal concentrations were used in the

409 figures. Also shown are the spearman correlation coefficients for each relationship.

410   The acidic nature of the aerosols raises the possibility that acid processing played a

411 major role in enhancing aerosol metal solubilities. During acid processing, acidic species have

412 to overcome the buffering capacity of the aqueous aerosol particle to raise the aerosol acidity

413 level to the point where the dissolution of metal species is thermodynamically favored. Since





sulfate is the main aqueous-phase acidic species in fine aerosols in Hong Kong, we first
analyzed the relationships between the concentrations of water-soluble metals and sulfate.
Figure 5 shows that despite the scatter in the datasets, the concentrations of water-soluble
metals were positively correlated with the concentration of sulfate, though the correlations
between the concentrations of sulfate and water-soluble Al and Ni were not statistically
significant. These positive correlations could be due, in part, to the water-soluble metals and
sulfate precursor (i.e., SO₂) being emitted from the same sources. However, the masses of
primary water-soluble aerosol metals are not known. The positive correlations could also be
due to the role that sulfate plays in aerosol metal dissolution during acid processing.

To investigate the role that sulfate played in controlling aerosol metal solubilities, we

analyzed the relationships between the metal fractional solubilities and sulfate concentration.
Analyses of the correlations between the metal fractional solubilities and sulfate concentration
(Table 1 and Figure 6) indicated that the Cr, Fe, Co, Cu, Pb, and Mn fractional solubilities were
positively correlated with the sulfate concentration, and these correlations were statistically
significant. This implied that sulfate played a key role in the formation of water-soluble Cr, Fe,
Co, Cu, Pb, and Mn, likely though sulfate-driven acid dissolution of their water-insoluble forms.
Conversely, the positive correlations between the sulfate concentration and the Al, Ni, V, and
Cd fractional solubilities were weak and not statistically significant. Interestingly, the V and
Cd fractional solubilities showed weak correlations with the sulfate concentration ($R = 0.14$
and $R = 0.04$, respectively), whereas their water-soluble concentrations showed strong
correlations with the sulfate concentration ($R = 0.65$ and $R = 0.74$, respectively). It is possible
that the strong correlations of sulfate concentration with water-soluble V and Cd concentrations
but not with V and Cd fractional solubilities were due to a large fraction of water-soluble V and
Cd having the same emission sources as the sulfate precursor (i.e., SO₂).
**Table 1:** Spearman rank correlations between the metal fractional solubilities and $W_i$ and $H_{air}^+$
in fine aerosols[a]

| Metal | Sulfate | $W_i$ | $H_{air}^+$ |
|:-----:|:-------:|:-----:|:-----------:|
| Cr | **0.62** | **0.42** | **0.48** |
| Al | 0.14 | 0.08 | 0.14 |





| | | | |
|---|---|---|---|
| Fe | **0.53** | **0.31** | **0.50** |
| Ni | 0.03 | 0.01 | 0.18 |
| Co | **0.41** | **0.41** | **0.23** |
| Cu | **0.74** | **0.72** | **0.24** |
| Pb | **0.53** | **0.41** | **0.34** |
| Mn | **0.49** | **0.43** | **0.23** |
| V | 0.14 | 0.01 | 0.21 |
| Cd | 0.04 | 0.10 | 0.13 |

[a] Bold: statistically significant ($p < 0.05$)

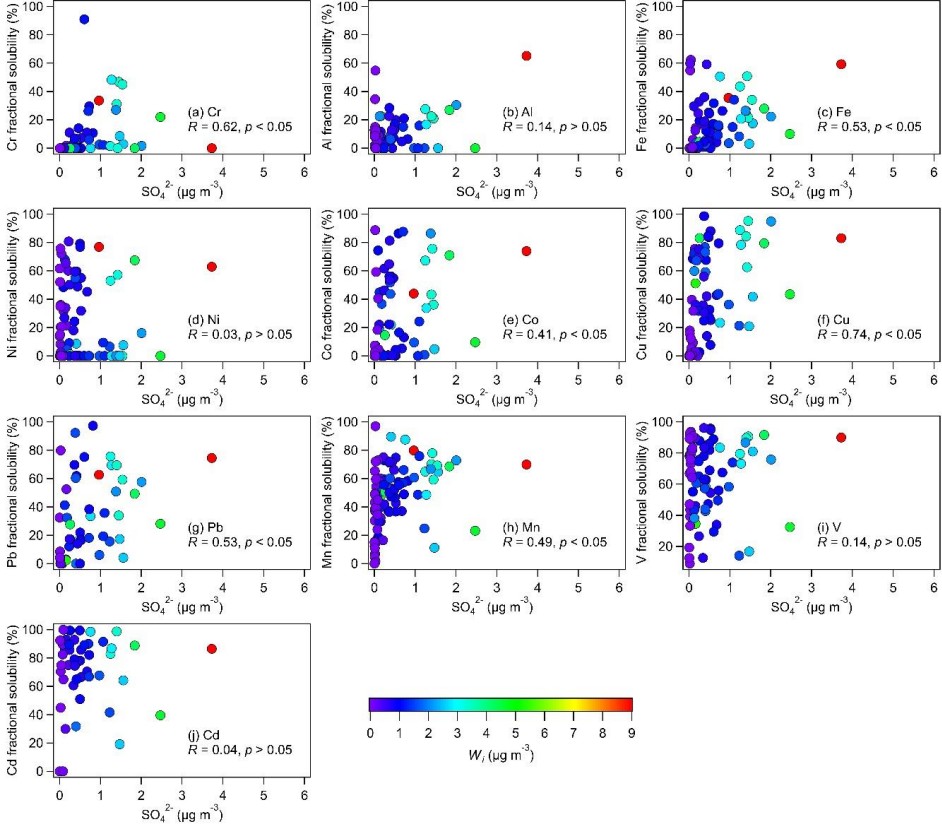


**Figure 6:** Relationships between the metal fractional solubilities and sulfate mass concentration in fine aerosols. Only data with non-zero total metal concentrations were used in the figures. Also shown are the spearman correlation coefficients for each relationship. The symbols are colored by the corresponding $W_i$ concentrations calculated by ISORROPIA-II. The $W_i$ concentrations increased with sulfate concentrations.



High levels of aerosol acidity and liquid water are generally needed for the acid
dissolution of metals in an aqueous aerosol particle. In addition to being the main contributor
to aerosol acidity levels (i.e., $H_{air}^+$), sulfate is a highly hygroscopic species that will influence
the overall aerosol water uptake behavior, which will drive $W_i$. Sulfate was the main driver of
$W_i$ in fine aerosols in our study since the mass concentrations of nitrate (another highly
hygroscopic species) were very low (about 18 times lower than sulfate, on average). Both $W_i$
and $H_{air}^+$ were controlled primarily by sulfate (sulfate and $W_i$ $R = 0.90$, $p < 0.05$; sulfate and
$H_{air}^+$ $R = 0.63$, $p < 0.05$). Thus, we analyzed the relationships between the aerosol metal
fractional solubilities and $W_i$ and $H_{air}^+$ (Figures S6 and S7). Table 1 shows that correlations
between the Al, Ni, V, and Cd fractional solubilities and $W_i$ and $H_{air}^+$ were weak. Together, the
weak correlations between the fractional solubilities of Al, Ni, V, and Cd and sulfate, $W_i$, and
$H_{air}^+$ implied that acid processing may have played a minor role in enhancing the solubilities
of these four metals. Other atmospheric processes beyond acid processing (e.g., cloud
processing, photoreduction) could have played more important roles in enhancing the
solubilities of these four metals (Zhu et al., 1993; Spokes et al., 1994; Kuma et al., 1995). It is
possible that these four metals had slow acid dissolution rates as a result of their mineralogy
and oxidation states. The impacts of mineralogy and oxidation states on the susceptibilities of
water-insoluble Al, Ni, V, and Cd to acid dissolution are currently not known. However,
previous studies showed that different aerosol Fe mineralogy and oxidation states have
different susceptibilities to acid dissolution that will occur at different timescales (Ingall et al.,
2018). Hence, analogous to Fe, it is possible that the mineralogy and oxidation states of Al, Ni,
V, and Cd in the collected aerosols may have resulted in these four metals being less susceptible
to acid processing, which in turn caused them to undergo slow sulfate-driven acid dissolution
from water-insoluble forms to water-soluble forms.
Table 1 shows that the Cr, Fe, Co, Cu, Pb, and Mn fractional solubilities were positively
correlated with $W_i$ and $H_{air}^+$, and these correlations were statistically significant. Together, the
strong correlations between the fractional solubilities of Cr, Fe, Co, Cu, Pb, and Mn and sulfate,
$W_i$, and $H_{air}^+$ indicated that acid processing likely played a major role in enhancing the
solubilities of these six metals. The fractional solubilities of Co, Cu, Pb, and Mn were more



strongly correlated with the $W_i$ concentration than with the $H_{air}^+$ concentration. This suggested
that $W_i$ had a stronger influence on the acid dissolution of Co, Cu, Pb, and Mn. The strong
influence that $W_i$ has on the metal fractional solubility could be explained by the role of aerosol
water as a reaction medium for the acid dissolution of metals in an aqueous aerosol particle.
Wong et al. (2020) previously showed that at a relatively constant aerosol pH, a decrease in $W_i$
will lead to a decrease in the reaction medium volume, which in turn will lead to decreases in
the overall formation rates of water-soluble metals. Conversely, the fractional solubilities of Cr
and Fe were more strongly correlated with the $H_{air}^+$ concentration than with the $W_i$
concentration. This suggested that the aerosol acidity levels had a stronger influence on the
acid dissolution of Cr and Fe.
Interestingly, variability in the aerosol pH did not appear to be a key driver of the
variability in the solubilities of Cr, Fe, Co, Cu, Pb, and Mn. It was difficult to discern aerosol
pH-dependent fractional solubility trends for these six metals, and their fractional solubilities
were not highly correlated with aerosol pH (Figure S8). This could be attributed partly to the
scatter in the datasets caused by differences in the metal solubilities and pH in individual
aerosol particles that would not be captured by the bulk chemical analysis and thermodynamic
modeling performed in this study. The absence of obvious aerosol pH-dependent fractional
solubility trends could also be due to the insensitivity of aerosol pH to the variability of sulfate
($R = -0.22$, $p < 0.05$). Based on Equation (1), the aerosol pH could be viewed simply as the
ratio of $H_{air}^+$ and $W_i$. Both $W_i$ and $H_{air}^+$ were highly variable in this study, and both were
controlled primarily by sulfate. As a result, the ratio of $H_{air}^+$ and $W_i$, or the aerosol pH, would
be fairly insensitive to sulfate even though it was driven primarily by sulfate. Previous studies
have similarly reported weak or the absence of aerosol pH-dependent metal fractional solubility
trends despite evidence of aerosol metal dissolution being enhanced by acid processing (Shi et
al., 2020; Wong et al., 2020).
**4. Conclusions**
In this study, we investigated the abundance and fractional solubilities of ten metals (Fe,
Cu, Al, V, Cr, Mn, Co, Ni, Cd, and Pb) in size-fractionated aerosols collected at an urban site

in Hong Kong. Weekly aerosol samples were collected for a month during different seasons from March 2021 to January 2022. The main objective of this study was to identify the key factors that controlled metal solubilities in fine aerosols, with a focus on aerosol metal dissolution via the acid processing and metal-organic complexation mechanisms. Hence, other aerosol chemical species were measured in addition to the total and water-soluble metals.

Higher mass concentrations of total metals were usually measured during the winter and/or spring seasons. This was likely due to the long-range transport of polluted air masses by northly prevailing winds from emission sources located in continental areas north of Hong Kong. The total metals could be arranged in the following order based on their abundances: Fe > Al > Cu > Mn > Pb > Ni > Cr > V > Co > Cd. This order of abundance was the same for both fine and coarse aerosols. The major sources of the total metals were sea salt, dust, ship emissions, and industrial activities. Higher mass concentrations of water-soluble metals were also usually measured during the winter and/or spring seasons. With the exception of Cu, the water-soluble metals had higher mass concentrations in fine aerosols than in coarse aerosols. The mass concentrations of water-soluble metals generally correlated with the mass concentrations of total metals, which implied that the water-soluble metals were largely derived from their total metals through atmospheric processing and/or that water-soluble and water-insoluble metals have the same emission sources. The study-averaged metal fractional solubilities spanned a wide range for both fine (8.8 % to 70.3 %) and coarse (1.4 % to 54.3 %) aerosols. With the exception of Cu and Co, the metals exhibited higher fractional solubilities in fine aerosols compared to coarse aerosols. The aerosol size-dependent metal fractional solubility could potentially be attributed to differences in the composition and metal mineralogy which resulted in different metal dissolution rates and/or mechanisms for aerosols of different sizes.

The fine aerosols collected in this study were mostly acidic, with about 60 % of the calculated pH values below 3. The acidic nature of the fine aerosols combined with oxalate (which forms metal-organic complexes easily) not being detected in our aerosol samples suggested that organic ligand-promoted dissolution likely played a minor role in enhancing aerosol metal solubilities. This is because organic ligand-promoted metal dissolution is a slow



process, and it plays a minor role in metal dissolution under low pH conditions. Our analyses
showed that sulfate, which is the dominant fine aerosol acidic species, exhibited statistically
significant positive correlations with both the water-soluble concentrations of Cr, Fe, Co, Cu,
Pb, and Mn and their fractional solubilities. In addition, sulfate controlled $W_i$ and $H_{air}^+$, both of
which are needed for acid dissolution of metals in an aqueous aerosol particle. The water-
soluble concentrations of Cr, Fe, Co, Cu, Pb, and Mn and their fractional solubilities exhibited
statistically significant positive correlations with both $W_i$ and $H_{air}^+$. Together, the strong
correlations between the fractional solubilities of Cr, Fe, Co, Cu, Pb, and Mn and sulfate, $W_i$,
and $H_{air}^+$ indicated that acid processing likely played a major role in enhancing the solubilities
of these six metals. The fractional solubilities of Co, Cu, Pb, and Mn were more strongly
correlated with the $W_i$ concentration than with the $H_{air}^+$ concentration, which implied that $W_i$
had a stronger influence on the acid dissolution of these four metals. The fractional solubilities
of Cr and Fe were more strongly correlated with the $H_{air}^+$ concentration than with the $W_i$
concentration, which implied that the aerosol acidity levels had a stronger influence on the acid
dissolution of these two metals. Conversely, our analyses suggested that acid processing played
a minor role in enhancing the solubilities of Al, Ni, V, and Cd. It is possible that the mineralogy
and oxidation states of these four metals made them less susceptible to acid processing.
In conclusion, this study highlights the key role that sulfate plays in controlling the
solubilities of a host of metals in fine aerosols (in this case, Cr, Fe, Co, Cu, Pb, and Mn). This
is mostly due to sulfate's ability to both strongly acidify the aerosol particle and provide the
liquid reaction medium needed for the acid dissolution of metals. Although this study was
performed at an urban site in Hong Kong, we expect our findings to broadly apply to other
urban areas in Hong Kong and South China, where sulfate is the dominant acidic and
hygroscopic component in fine aerosols. Results from this study can also provide insights into
how the solubilities of different aerosol metals will change with the decrease in sulfate as Hong
Kong and other cities in South China transition away from coal combustion as their main
energy source to improve local and regional air quality and combat climate change.
**Data availability:** The data used in this publication is available to the community and can be
accessed at: https://doi.org/10.5281/zenodo.7013770 (Yang et al., 2022).



**Author contributions:** J.Y. and T.N. designed the study. J.Y. collected the field samples. J.Y., L.M., and W.C.A. performed chemical analysis of the field samples. J.Y., X.H., Y.M., and T.N. analyzed the data. J.Y. and T.N. prepared the manuscript with contributions from all co-authors.

**Competing interests:** One of the authors is a member of the editorial board of *Atmospheric Chemistry and Physics*. The peer-review process was guided by an independent editor, and the authors also have no other competing interests to declare.

**Acknowledgements:** This work was supported by the Research Grants Council of Hong Kong (project number 21304919).

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
