# Peer review of "Measurement Report: Abundance and fractional solubilities of aerosol metals in urban"

_Atmospheric Chemistry and Physics, 2022_

## Author Comment (AC3)

We greatly value the careful reading and the detailed comments provided by the referees. The responses to the comments of the referees in our direct reply (shown below) and within the revised manuscript (see marked copy) are provided below. The pages and lines indicated below correspond to those in the marked copy.

**Response to Referee 1 (Referees' comments are italicized)**

1. Referee comment: "Line 174-178: After sonication, how were insoluble materials separated from the aqueous extracts? Was it achieved by high speed vortexing? Filtration is usually used by many studies, with filter pore-size clearly stated. More information should be provided here."

**Author response:** Both high speed vortexting and filtration were used for aqueous extraction. Filtration was performed using 0.22 µm pore size nylon filters (Jinteng Instrument Co., Tianjin, China). This information has been included in the revised manuscript:

Page 6 line 167: "One of the four pieces was extracted in purified deionized water via sonication (1 hour), followed by high speed vortexing at 3000 rpm (15 minutes). The resulting aqueous extract was filtered using 0.22 µm pore size nylon filters (Jinteng Instrument Co., Tianjin, China) before it was analyzed by a Total Organic Carbon (TOC) analyzer (TOC-VCSH, Shimadzu, Japan) to determine the concentration of water-soluble organic carbon (WSOC)."

Page 6 line 172: "The second filter piece was similarly extracted in purified deionized water via sonication and high speed vortexing at 3000 rpm, and filtered using 0.22  $\mu$ m pore size nylon filters before it was analyzed by an ion chromatography (IC) system (Dionex ICS-1100, ThermoFisher Scientific, USA) using an isocratic method to determine the concentrations of water-soluble anions (NO3-, SO42-, Cl-, and C2O42-) and cations (NH4+, Na+, K+, Ca2+, and Mg2+)."

Page 6 line 189: "One filter piece was extracted with purified deionized water in metalfree centrifuge tubes via sonication (1 hour), followed by high speed vortexing at 3000 rpm (15 minutes). The resulting aqueous extract was filtered using 0.22  $\mu$ m pore size nylon filters, and acidified with 2 % HNO3 prior to storage at 4 °C before chemical analysis of water-soluble metals."

Page 7 line 218: "The resulting solution was filtered using 0.22  $\mu$ m pore size nylon filters, and then stored at 4 °C before chemical analysis of total metals."

2. Referee comment: "Line 250-259: Can the authors show and discuss correlations between *Al* and other metals? This may give further insights (in addition to size distribution) to their sources."

**Author response:** As requested, we have added a figure and a discussion about the correlations between Al and other metals to the revised manuscript:

Page 10 line 291: "The mass concentrations of the two most abundant metals, Fe and Al,

were usually higher than 10 ng m-3 in both fine and coarse aerosols. Fe, Al, and Cu had substantially higher mass concentrations in coarse aerosols than in fine aerosols. The positive correlations between the mass concentrations of Al with the mass concentrations of Fe and Cu were the strongest among the nine metals (R = 0.62 and R = 0.52, respectively) and statistically significant (Figure S3), which could be explained by large mass concentrations of Al, Fe, and Cu originating from similar sources. These three metals are known to originate mainly from dust sources (e.g., mineral dust and road dust) (Hopke et al., 1980; Garg et al., 2000; Adachi and Tainosho, 2004; Lough et al., 2005; Chow et al., 2022). This is consistent with results from our PMF source apportionment analysis, which showed that the "dust" factor had large mass contributions from Al, Fe, and Cu (Figures S4 and S5). Mn, Ni, V, and Cd had higher mass concentrations in fine aerosols than in coarse aerosols. These four metals are known to be consistently found in aerosols from anthropogenic sources such as vehicle and ship emissions, combustion and industrial processes (Chow et al., 2022). Pb, Cr, and Co had mostly similar concentrations in the fine and coarse aerosols. Interestingly, the mass concentrations of Mn and Cr were positively correlated with the mass concentration of Al (R = 0.42 and R = 0.33, respectively), and these correlations were statistically significant (Figure S3). Our PMF analysis apportioned Al to two factors, "dust" and "industrial factor 1", though the Al contribution to "industrial factor 1" was substantially smaller compared to "dust" (Figures S4 and S5). The "dust" factor had a significant Mn contribution, which could explain the strong correlation between the mass concentrations of Al and Mn. Cr was apportioned primarily to "industrial factor 1", which could explain the strong correlation between the mass concentrations of Al and Cr. The mass concentrations of Ni, V, Cd, Pb, and Co showed weak correlations with the mass concentration of Al (Figure S3)."

Figure S3: Relationships between the mass concentrations of total Al and the other total metals

in coarse and fine aerosols. Only data with non-zero total metal concentrations were used in the figures. Also shown are the spearman correlation coefficients for each relationship.

3. Referee comment: "Line 352-353: Several previous studies, including our work (Zhang et al., 2022) and references therein, also found that Fe solubility (as also other metals) was higher in fine particles than coarse particles. The authors may consider discussing these studies."

Author response: We thank the referee for bringing these previous studies to our attention. We have added the references to our revised manuscript.

4. Referee comment: "Line 392-405: As this work did not manage to detect oxalate in aerosol particles, I feel this paragraph is tedious and not very relevant. The authors may consider making it more concise."

Author response: As requested, we have made this paragraph more concise in the revised manuscript:

Page 17 line 484: "To investigate whether organic ligands influenced aerosol metal solubilities in this study, we attempted to measure oxalate in the size-fractionated aerosol samples using IC. However, we could not detect oxalate, which indicated that the concentrations of oxalate (if present) were below the detection limits of our IC instrument. It should be noted that although a recent study reported the copresence of Fe and oxalate in individual aerosol particles at a suburban site in Hong Kong using single particle mass spectrometry (Zhou et al., 2020), organic ligand-promoted metal dissolution is a slow process, and it plays a minor role in metal dissolution under low pH conditions (Zhu et al., 1993). The fine aerosols collected in this study were mostly acidic, with about 60 % of the calculated pH values being less than 3. This suggested that organic ligand-promoted dissolution may have played a minor role in enhancing aerosol metal solubilities in this study due to the acidic nature of the aerosols."

5. Referee comment: "Line 67-69: The first sentence only mentioned "many atmospheric processes", but the second and third sentences mainly discussed ocean biogeochemistry. The authors may need to modify the first sentence to make it more appropriate."

Author response: The following changes have been made to the revised manuscript:

**Page 3 line 77: "Water-soluble metals also play important roles in ocean biogeochemistry and atmospheric processes."**

6. Referee comment: "Line 203-204: would it be enough to use only one unit (ng/L) here?"

Author response: We have made the requested changes in the revised manuscript.

7. Referee comment: "*Line 239: It may be better to use particle diameter for x-axis in Figure 1.*"

**Author response:** As requested, the x axes in Figures 1 and 3 has been changed in the revised manuscript. To make the colors in Figures 1 and 3 more discernible to readers with color vision deficiencies, we also changed the color scheme for these two figures. It should be noted that

the incorrect set of data was plotted for Cu in Figure 3 of the original manuscript (we previously accidentally plotted the Co data instead of the Cu data in panel 3f). This has been corrected in the revised manuscript. The corrected figure does not change the conclusions drawn in this study. The following changes have been made to the revised manuscript: